## Research Article

health system building blocks; mental health planning; mental health reform; mental health systems strengthening; participatory approaches

**Corresponding author:**
Giovanni Sala;
Email: gio.sala@outlook.com

# Comprehensive planning for mental health system reform: lessons learned from the WHO Special Initiative for Mental Health

Giovanni Sala 🄳 and Alison Schafer 🄳

Department of Mental Health, Brain Health and Substance Use, World Health Organization, Geneva, Switzerland

## Abstract

The WHO Special Initiative for Mental Health is an ambitious cross-country program currently operating in nine countries: Argentina, Bangladesh, Ghana, Jordan, Nepal, Paraguay, Philippines, Ukraine and Zimbabwe. The goal of the program is to promote mental health system reform at the national level, shifting from an institutional care model to community-based care following a person-centered and rights-based approach. An initial planning phase of the program involved developing country-specific implementation plans through multi-stakeholder workshops, which resulted in national-level logical frameworks. Through the present study, a thematic analysis was carried out to explore the main commonalities across all countries' plans. The results show that countries converge on a number of commonalities, including the broad-reaching, multifaceted and multi-sectoral nature of national reform strategies, a focus on person-centered and community-driven initiatives and recognition of the added value of institutional structures and expert advice on key issues such as monitoring and evaluation. The results of the present study can help guide future exercises of this kind in other countries.

## Impact statement

This article presents significant insights into the global effort to reform mental health systems. By examining commonalities and differences across nine diverse countries, this research underscores the importance of holistic and inclusive approaches in mental health reform. The wider impact of this research is its ability to guide other countries in designing and implementing mental health reforms that address the unique needs of their populations while aligning with international best practices. The focus on human rights, person-centered care and community-driven initiatives not only promotes equitable access to mental health services but also emphasizes the role of people with lived experience in their design, implementation and evaluation. By fostering collaboration across sectors and involving a broad range of stakeholders, this initiative lays the groundwork for sustainable mental health systems globally. Moreover, the research contributes to the growing body of evidence supporting the necessity of integrating mental health into primary care and adopting multi-sectoral strategies. These findings are particularly relevant for countries seeking to transition from institutional to community-based care, offering lessons learned from other countries for implementing long-lasting reforms that align with human rights principles and improve the lives of those affected by mental health conditions.

## Background

Despite the efforts of mental health workers, advocates and researchers worldwide over recent decades – and despite anecdotal experiences that more people acknowledge the importance of mental health – there remain limited and isolated examples of successful national-level mental health system development, with the treatment gap for mental health conditions remaining extremely high (Thornicroft et al., 2017).

World Health Organization (WHO) Special Initiative for Mental Health: Universal Health Coverage for Mental Health is an ambitious program designed to increase access to mental health care for 100 million more people across 12 countries. Though many aspects of mental health systems are essential, the WHO opted for the Special Initiative for Mental Health to focus on two areas of action in each country: (i) advancing mental health policies, advocacy and human rights and (ii) scaling up interventions and services across community-based, general health and specialist settings. What is different about the Special Initiative for Mental Health, as opposed to other mental health programs, is its focus on mental health system reform through transformation of mental health systems, including work not only on mental health services but also on other health system building blocks, such as human resources, financing, governance, information management and access to medicines (WHO, 2007).

The initiative was launched in 2019 and began implementation in 2020 with six early adopter countries: Bangladesh, Jordan, Paraguay, Philippines, Ukraine and Zimbabwe. Nepal, Ghana and Argentina saw the Initiative expand in the following years. Countries participating in the WHO Special Initiative for Mental Health were mainly selected according to demonstrable government commitment (and possibly early progress) to mental health services, written agreement from ministries of health to engage with WHO for mental health systems strengthening for 5 years, availability of human resources in ministries of health to actively collaborate with available human resources in the WHO country office, government willingness to be part of a global program for learning, engaged local partners for joint implementation, financial resources (local, regional and international) and community needs for scaling mental health services, including countries seeking to shift from facility-based to community-based mental health services.

Deinstitutionalization, which entails shifting the focus of mental health care from large, centralized psychiatric institutions to community-based services, is a cornerstone of sustainable mental health system reform (WHO, 2023a). Countries that had already expressed some political will and initiated steps toward this transition were considered to be better positioned to collaborate on scaling up comprehensive, person-centered care. Despite the relatively modest financial investment, the initiative has achieved substantial system-level changes that have paved the way for improved access and service quality. With total expenditures amounting to USD 25 million, the initiative has provided access to newly available community-based mental health services for over 50 million people (WHO, 2024; Kestel et al., 2025). Remarkably, this translates to a cost of less than USD 0.50 per person newly reached – a testament to the initiative's cost-efficiency and strategic impact. These impressive returns on investment highlight the importance of sustained financial support for long-term mental health programs, even as such funding remains challenging to secure.

The WHO Special Initiative for Mental Health's work is intended to be structured similarly across countries, with adaptations based on context and needs. Each country follows a typical project management cycle, as described in Figure 1.

As countries progress through the implementation phase of the WHO Special Initiative for Mental Health, lessons are emerging and important to document. This paper looks to build upon a previous publication detailing the results of the Situational Assessment phase across countries (Kemp et al., 2022), by focusing on the Planning phase to summarize the lessons learned and the main commonalities between each country's project design documents.

### WHO Special Initiative for Mental Health country design consultations

A common approach was taken to developing program designs across countries – in the form of logical frameworks (or logframes in short) (Team Technologies, Middleburg, Virginia, 2005) – with a common methodology being developed and adapted to local realities and COVID-19 restrictions.

Prior to the COVID-19 pandemic, Ukraine benefited from a face-to-face 2-day multi-stakeholder workshop. Thereafter, Bangladesh, Jordan, Philippines, Paraguay and Zimbabwe completed design consultations online. As COVID-19 risks eased, Ghana, Nepal and Argentina were able to undertake face-to-face workshops again.

Over time and through multiple phases, a final draft logframe was validated by all members. This draft logframe was shared with WHO Regional and Headquarters technical staff, with inputs incorporated after an additional round of consultation (where needed). The final proposed logframe was presented to representatives of the country's Ministry of Health for official validation and agreement to proceed toward implementation.

The Special Initiative's design consultation processes were designed to be as inclusive and iterative as possible, engaging a

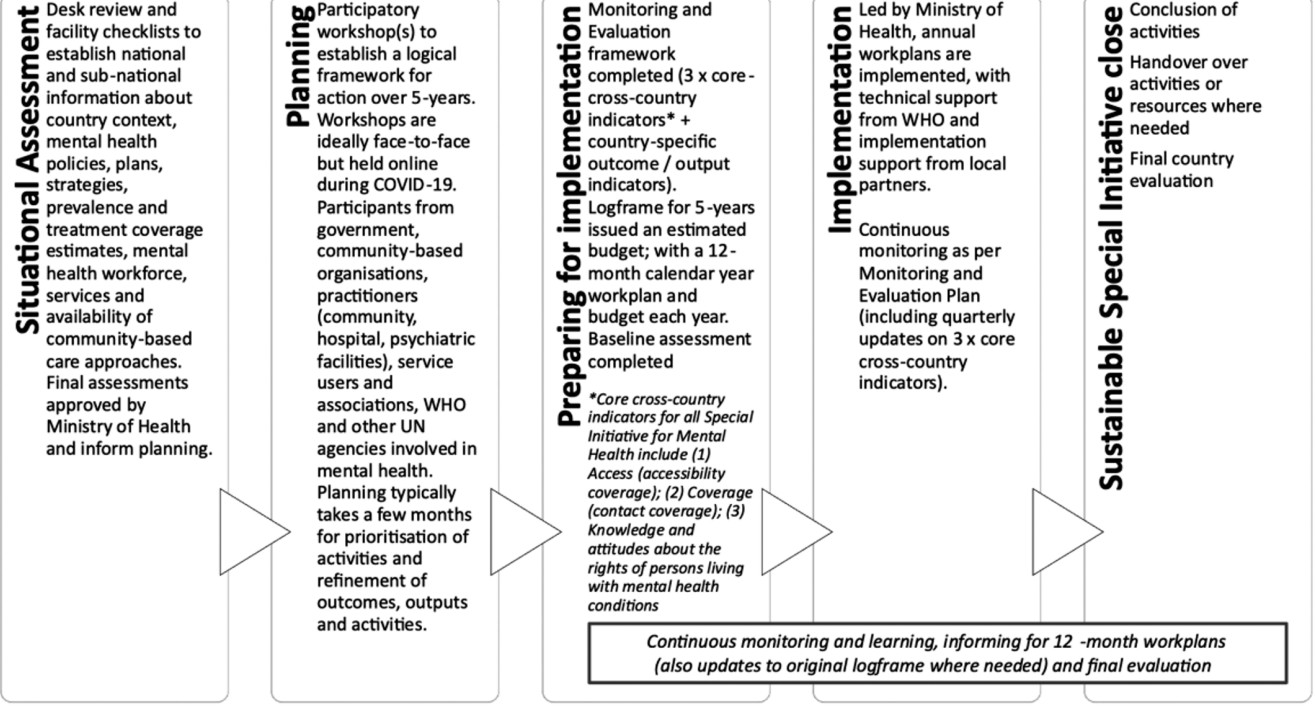

**Figure 1.** Description of the WHO Special Initiative for Mental Health's typical project management cycle.

wide variety of stakeholders and collecting their feedback at different stages of the process.

The objective of this study was to identify similarities and differences across Special Initiative country logframes. The results may assist other countries to undertake similar prioritization and planning exercises. It may also be relevant for national and/or sub-national-level planning, particularly where demands relate to mental health systems strengthening approaches.

## Methods

### Materials

Analyses focused on the information and data contained in the WHO Special Initiative for Mental Health logframes developed in Argentina, Bangladesh, Ghana, Jordan, Nepal, Paraguay, Philippines, Ukraine and Zimbabwe between March 2020 and October 2022. The objects of analysis limited itself to Outcome and Output statements within each logframe, not the Activities nor the Indicators. Activities were deemed to be too detailed for this study, and they would not have added significant information to the overall analysis on top of that retrieved from Outcomes and Outputs. Indicators were deemed to be of a completely different nature and hence should be studied through a separate study and publication focusing on monitoring and evaluation mechanisms for mental health system reforms.

### Participants

The number of participants engaged in the logframe design process varied widely across countries. Nepal, Paraguay and Ukraine comprised consultation groups of between 25 and 40 participants, whereas the Philippines and Zimbabwe availed the opportunity for online consultations to include many more people, often engaging over 100 participants per consultation session. Irrespective of the total number of participants, all consultations included, as much as possible, a wide variety of stakeholders: mental health policymakers, practitioners, people with lived experience of mental health conditions, service managers, local academia, people from scientific societies, community-based organizations, NGOs, UN agencies and government representatives. Participants were also from multiple sectors of work, such as health, education, social services, gender, justice, HIV/AIDS and private sector programs. Consultations were led by the representatives of the Ministry of Health of each country, supported by WHO staff throughout.

### Analysis

Data were analyzed using a Thematic Analysis approach, as described in Braun and Clarke (2006). The process spanned more than 1 year, as different country logframes became available at different time points. This long span of time allowed for a recursive process through which themes were double-checked multiple times against the available information and data contained in each country's logframe.

Although most of the data were already known to the researchers, an active effort was made to take an inductive approach in analysis, keeping an open mind to patterns and potential themes arising from the data.

After an initial phase of familiarization with the data (i.e., Outcome and Output statements from logframes), a coding phase captured the main elements. Once all statements were coded, the codes were combined into 29 themes. Through further analysis and multiple reviews of the data, a thematic map was created. This map grouped the 29 themes together into five higher-level themes, each with its own sub-themes and interconnections.

See Figure 2 for a map of all codes and themes. Each statement could have as little as one or as many as four codes, depending on how broad it was, for a total of 407 different codes.

## Results

The high-level themes identified through this analysis were linked to (a) the approaches and overarching principles that guided logframe

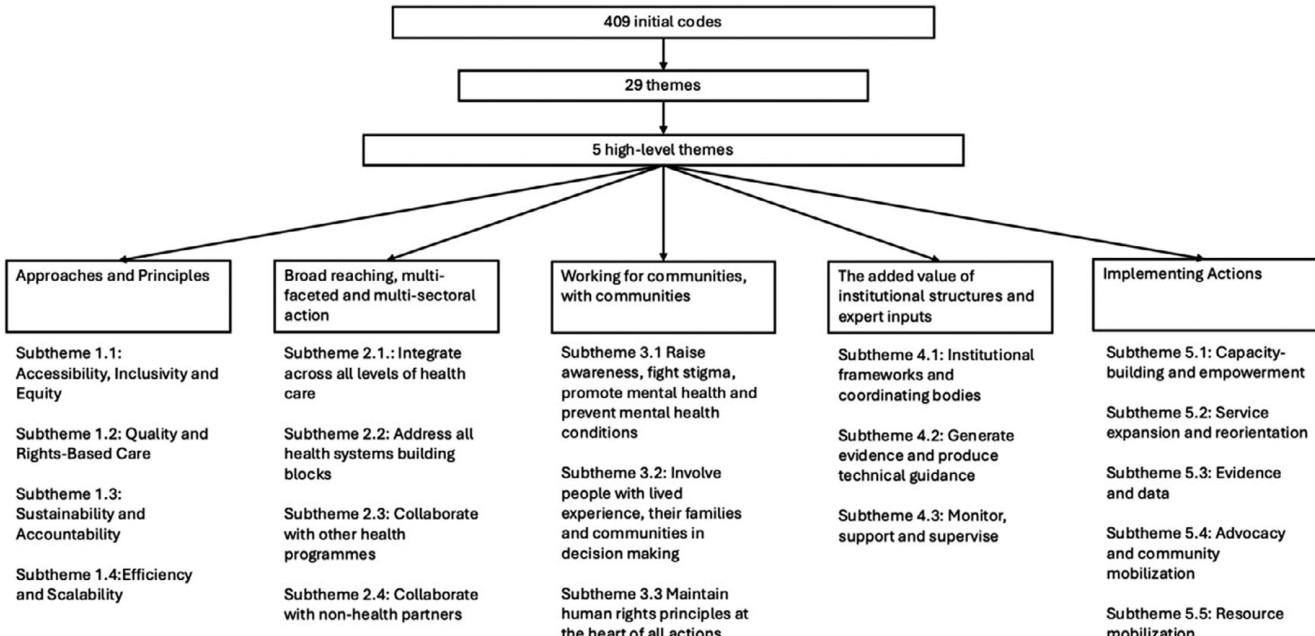

**Figure 2.** Code map of all codes, themes, high-level themes and respective subthemes.

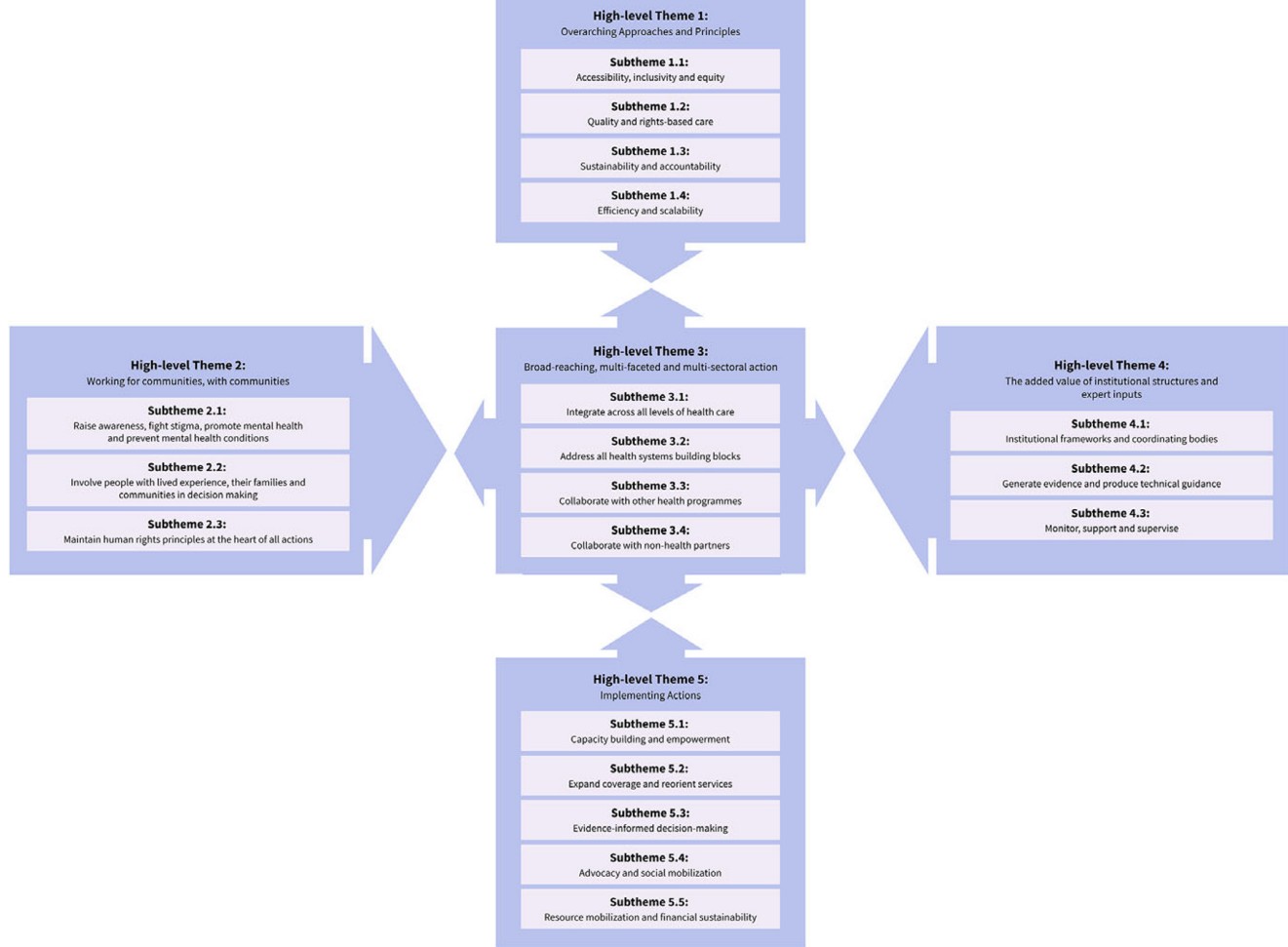

**Figure 3.** Pictorial summary of main themes and subthemes.

development, (b) the goals for mental health system reform and (c) the general strategies identified for achieving those goals. These themes, and their corresponding subthemes, are visually summarized in Figure 3, which provides a pictorial overview of the thematic structure derived from the analysis.

### *High-level theme 1: Overarching approaches and principles*

The thematic analysis of principles embedded in the logframes reveals a coherent set of overarching values that guide how mental health system reforms were designed. These principles serve as ethical and operational anchors, ensuring that the proposed actions remain inclusive, sustainable and rights-based.

A prominent guiding principle was **accessibility, inclusivity and equity** (Subtheme 1.1), which emphasizes the need for mental health services to be physically and financially accessible to all, regardless of socioeconomic status, age or location.

**Quality and rights-based care** (Subtheme 1.2) was another recurring principle, highlighting countries' commitment to delivering evidence-based services that uphold dignity, autonomy and non-discrimination. This reflects an understanding that mental health systems must not only be effective but also rooted in human rights protections to ensure respectful and person-centered care.

**Sustainability and accountability** (Subtheme 1.3) further guide implementation, emphasizing the need for enduring mental health reforms supported by robust governance and financial frameworks. Accountability mechanisms are also prioritized to foster transparency, track progress and ensure that reforms remain aligned with their intended goals.

Finally, **efficiency and scalability** (Subtheme 1.4) underline the necessity of building systems capable of expanding service provision to meet increasing demands.

### *High-level theme 2: Broad-reaching, multifaceted and multi-sectoral action*

Countries planned for actions in the most holistic way possible. Broad-reaching interventions were planned to reach as many people as possible, often specifying vulnerable subgroups in an exhaustive manner. Multi-faceted strategies were often applied to tackle mental health issues from different angles, and multi-sectoral action was recognized by all countries as a necessary strategy to respond to the complex mental health needs of the population.

#### *Subtheme 2.1: Integrate across all levels of health care*

All countries emphasized the need to develop and expand community-based mental health services, which included, as a matter of priority,

the integration of mental health services as part of primary health care. Strengthening acute secondary care to treat those in need of specialist services, as well as to provide supervision for non-specialist service providers in primary and community settings, was also highlighted. All countries identified the need to move away from tertiary, facility-based care provided in psychiatric hospitals or institutions. Moreover, countries converged on the idea that all levels of the health system need to be involved for real transformation to take place, from national to local levels. For example, Ghana's logframe reflected this need through the following output:

- *Ghana Output 3.2: Well-defined mental health services framework developed for services to be delivered at each level of care.*

### Subtheme 2.2: Address all health systems building blocks
Country designs highlighted the importance of mental health systems needing to improve on all identified 'building blocks' commonly used when working on health systems strengthening functions, including health services, human resources, financing, governance, information management and medicines. Examples include:

- *Zimbabwe Output 2.1: The mental health financing system is strengthened to provide affordable services across the country, with a focus on access and sustainability.*
- *Bangladesh Output 4.2: Technical assistance provided to develop and implement a human resource plan for mental health for central and district levels, including job descriptions, and plan for retention.*

### Subtheme 2.3: Collaborate with other health programs
Logframes reflected the need to collaborate with colleagues from other health sectors, such as communicable and noncommunicable diseases, women's and children's health, Universal Health Coverage, social determinants of health and emergency response, among others.
   Examples of collaboration with other health programs included:

- *Nepal Output 4.6: Mental health is integrated within other priority areas, including communicable and non-communicable diseases, public health and social protection programs.*

### Subtheme 2.4: Collaborate with non-health partners
Collaboration with entities outside of the ministries of health additionally featured as a priority for multiple countries. Other ministries, including those responsible for social services, employment, human rights, indigenous affairs, protection, education and other types of actors such as civil society, representatives of people with lived experience, academic institutions, scientific societies and the private sector, were thought necessary because they play a critical role in supporting community-based mental healthcare support services.

- *Paraguay Output 4.1: Access to social and economic support programs for persons with psychosocial disabilities is promoted through the establishment of strategic alliances with institutions in the public sector, the private sector and civil society.*

## High-level theme 3: Working for communities, with communities

Community-based approaches were very often prioritized through the logframes. This stemmed from the need to bring services closer to the population, as well as to put individuals at the center of the care system and improve the way their needs are being met. Of importance, the respect of human rights, the inclusion of people with lived experience and the increase in service demand.

### Subtheme 3.1: Raise awareness, fight stigma, promote mental health and prevent mental health conditions
Awareness raising and stigma reduction were featured in multiple country logframes as strategies for increasing access to services.
   Moreover, prevention and promotion activities were identified as key actions to contribute to improving populations' mental health and complement other strategies (Le et al., 2021).

- *Paraguay Output 4.3: Civil society is strengthened to lead activities aimed at reducing stigma and discrimination.*

### Subtheme 3.2: Involve people with lived experience, their families and communities in decision-making
The meaningful involvement of people with lived experience and civil society in design development processes is a priority for the WHO Special Initiative for Mental Health. Therefore, people with lived experience and/or families or organizations that represent people with lived experience helped ensure that they would also be included in all decision-making about mental health, including for planning, implementation and evaluation activities.

- *Ghana Output 1.6: Service users are organized and coordinated to deliver advocacy and representation on National, Regional and District structures (i.e., Tribunals, committees and groups).*

### Subtheme 3.3: Maintain human rights principles at the heart of all actions
Another result of co-creating designs with the participation of people with lived experience and civil society was the strong focus on human rights. All Special Initiative countries mentioned human rights in some way, albeit with varying degrees of emphasis. WHO's QualityRights package was commonly cited as a tool for working toward the improvement of human rights standards across services (WHO, 2019c).

- *Ukraine Outcome 2: The human rights of people with intellectual and psychosocial disabilities are promoted and included in relevant policies for mental health services.*
- *Bangladesh Output 1.5: Enabling environment strengthened for ensuring rights of those affected by mental health and psychosocial conditions throughout health and social services aligned to the Convention on the Rights of Persons with Disabilities (CRPD) and other WHO guidelines, such as QualityRights, adapted to suit the country context.*

## High-level theme 4: The added value of institutional structures and expert inputs

All countries recognized the role that government representatives at different levels, technical experts, researchers and decision-

makers play in mental health systems reform. The added value of their work is linked to the power of institutional frameworks and coordinating bodies, the use of evidence and guidelines to steer action and the importance of having high-level structures to monitor, evaluate and supervise the implementation of activities.

### Subtheme 4.1: Institutional frameworks and coordinating bodies

Institutional frameworks are conceived as laws, policies, strategies and action plans, and in some countries, more specific documents related to financing systems and human resource management. The fact that all countries mentioned the need to develop or review them in their logframes testifies to their importance and a worrying signal of their absence (or outdatedness).

Moreover, various types of coordinating and/or decision-making bodies and activities were included in country logframes, based on a shared conception that these should play a key role in accelerating action and advocating for mental health services. For example, the Philippines planned for a "*Fully functional Philippine Council for Mental Health (PCMH)*" *(Philippines Output 1.1)*.

### Subtheme 4.2: Generate evidence and produce technical guidance

Many countries agreed on the importance of having expert groups and researchers to provide guidance for the implementation of activities under the WHO Special Initiative for Mental Health. This reflects the need for continuing the generation of evidence and creation of standards and guidelines to guide field work. For example, Ukraine called for "*mental health service practice standards and indicators [to be] updated to optimise quality assurance and decision making*" *(Ukraine Output 6.1)*, and Argentina planned to "*design guidelines to accompany human resources in the provision of services*" *(Argentina Output 2.3)*.

### Subtheme 4.3: Monitor, support and supervise

Monitoring and supervision were viewed as another responsibility of experts and researchers, as a strong need for accountability was shared across groups. Most countries called for comprehensive monitoring and evaluation systems to be created at the national level, and for said systems to function at all levels of care. Paraguay, for instance, determined that a specific mechanism should be created, calling for a Mental Health Observatory for the "*monitoring and evaluation of laws, policies and action plans*" *(Paraguay Output 1.7)*. Others, such as Zimbabwe, highlighted the need for supervision of human resources, through "*the development and implementation of a system for compliance checks by the regulators to enhance quality of the MH workforce as well as ensuring that adequate facilities and structures are provided thereof*" *(Zimbabwe Output 1.5)*.

### High-level theme 5: Implementing actions

In addition to outlining what each country aims to achieve, the logframes offer valuable insights into the processes proposed to reach these goals.

A central process identified is **capacity building and empowerment** (Subtheme 5.1), which involves equipping frontline workers and decision-makers with the necessary skills and tools to provide quality, person-centered care. This includes training, supervision and the development of standardized guidelines and protocols to support service delivery.

Efforts to **expand coverage and reorient services** (Subtheme 5.2) are at the heart of the reform plans, with countries proposing the integration of mental health care into primary health services and the scaling-up of community-based interventions. This reorientation aims to reduce reliance on tertiary institutions and foster a shift toward accessible, localized care.

Another key approach is **evidence-informed decision-making** (Subtheme 5.3), underscoring the importance of data collection, research and information management systems. By using evidence to guide planning and policy adjustments, countries aim to ensure that reforms remain adaptive and impactful.

**Advocacy and social mobilization** (Subtheme 5.4) were identified as essential for building public trust, reducing stigma and fostering service demand. Public awareness campaigns, partnerships with civil society and the meaningful involvement of people with lived experience are viewed as key to ensuring that mental health becomes a societal priority.

Finally, to support these processes, **resource mobilization and financial sustainability** (Subtheme 5.5) are prioritized, with plans detailing actions to secure funding from diverse sources, strengthen financial management systems and allocate resources effectively.

### Differences between country logframes

While having commonalities, each country's logframe reflects its unique priorities and strategies based on specific contexts and needs.

For instance, Jordan and Ukraine prioritize human rights in their logframes, reflecting their sociopolitical landscapes and recent conflicts. Jordan emphasizes ensuring the rights of people with mental health conditions through policy and legal reforms. This is especially relevant, given the significant number of refugees in Jordan, which complicates its mental health landscape and requires strong legal frameworks to ensure universal health coverage.

On the other hand, Ukraine focuses on integrating human rights into mental health service provision, particularly in conflict-affected areas.

Bangladesh and Zimbabwe put particular emphasis on decentralizing human resources and task-sharing, given their demographic and socio-economic constraints and challenges in providing specialized care for large parts of the population. Bangladesh focuses on strengthening technical assistance for human resource development, whereas Zimbabwe emphasizes mental health financing to ensure sustainable service provision.

The Philippines emphasizes integrating mental health into various health programs, whereas Paraguay focuses on forming strategic alliances for social and economic support programs, accentuating broader social determinants of mental health.

On the other hand, Argentina's logframe reflects its relatively well-resourced mental health system and stronger government services, but also its reliance on tertiary care. The reform strategy emphasizes deinstitutionalization through strengthened local care networks and cross-sector collaboration, balancing innovation with established systems.

Nepal's logframe emphasizes the integration of mental health services into primary health care, building on the country's long-standing experience with this approach. Recognizing donor priorities, the strategy also aligns mental health initiatives with non-communicable disease programs to maximize resource efficiency and ensure sustained support. This dual focus leverages existing health infrastructure while expanding mental health care within broader public health efforts.

Finally, Ghana's logframe placed a strong emphasis on leveraging its prior experience in implementing QualityRights to embed human rights-based practices across mental health services and foster greater accountability and inclusion. By strengthening partnerships with advocacy groups and community stakeholders, Ghana aimed to create a more participatory and rights-oriented mental health system.

## Discussion

The objective of this study was to identify the commonalities between different Special Initiative for Mental Health country workplans and to define key elements to be considered by other countries when going through similar processes of planning for mental health system development or reform. The data showed that the strategies developed by different countries had many clear similarities in terms of their main areas of work and priorities identified, as well as a few differences.

The results show that all countries developed highly ambitious strategies that looked to be holistic and cover all types of populations in need while approaching mental health system strengthening from various angles. All country plans included both 'expert-driven' and 'community-driven' approaches, highlighting the importance of having person-centered, community-based initiatives while recognizing the important role that the political and technical spheres can play in mental health system strengthening, particularly through advocacy, supervision and generation of evidence.

The broad-reaching, multifaceted and multi-sectoral approach that countries took was exemplified by the wide range of strategies identified for mental health system strengthening, which included different pathways for action, and by the variety of partners that were sought for implementation of activities. This clearly signifies the intersectoral nature of mental health issues, as well as the range of opportunities for collaboration as a means to increase access to mental health services.

The "community-based approach" was mainly exemplified by actions related to the meaningful involvement of people with lived experience both in planning and evaluation of services, to the importance of raising awareness and fighting stigma related to mental health to increase demand for services, and to protect and promote the human rights of people living with mental health conditions.

It is extremely important for health system strengthening actions to be based on the meaningful involvement of people with lived experience (WHO, 2019a, 2019b, 2021), especially for the mental health sector, in which many service users have suffered human rights abuses, particularly in institutions (Drew et al., 2011). People with lived experience can be powerful advocates for rights-based and person-centered care and can give valuable contributions to both high-level documents, such as laws, policies and plans (WHO, 2023a), and technical resources describing how services function. It is important for people with lived experience to also be involved in the monitoring and evaluation of both national frameworks and services.

Human rights principles must be at the heart of all planning and execution of activities, as outlined in the CRPD (United Nations, 2006) and other international human rights conventions.

All countries participating in the Special Initiative for Mental Health included elements of mental health system reform and deinstitutionalization within their logframes, reflecting their critical role in achieving long-lasting improvements in mental health care. Deinstitutionalization facilitates the reorganization of mental health services by shifting care closer to where people live, ensuring it is accessible, respectful of human rights and responsive to a broad spectrum of needs. By embedding this reform in their priorities, countries acknowledged that sustainable mental health systems require not only the establishment of new services but also an ecosystem of support structures – ranging from governance and financing to workforce capacity – that enable effective, community-driven care.

Although stigma reduction is not one of the priorities of the WHO Special Initiative for Mental Health, various countries identified it as an important element to complement the creation or reconfiguration of services.

The "expert-driven approach" of logframes was mainly expressed through the need to create frameworks that reflect the latest global evidence and rights-based approaches for mental health system reform. This includes giving mental health political and legal status, supporting leadership and governance structures to fulfill their roles and investing in the generation of evidence for the development of technical tools for implementation.

The capacity of leadership and governance in mental health was identified as a gap by most countries, reflecting what had already been documented in the first version of the Comprehensive Mental Health Action Plan (2013–2020) (WHO, 2013). This need was still evident for the Special Initiative for Mental Health countries. Such sentiment is consistent with one of the three pathways to transformation put forward in the World Mental Health Report from 10 years later (WHO, 2022): to "Deepen the value and commitment" that is given to mental health, including through "committed leadership." However, the role of leadership was attributed not only to the political realm but also to the sphere of services. Supervision from experts and the structured health system was a top priority to ensure implementation and quality of care.

The thematic analysis highlights not only what countries aim to achieve but also how they plan to do so. The guiding principles emphasize accessibility, quality, equity, integration, sustainability and accountability, reflecting a commitment to person-centered, rights-based care. By embedding comprehensive mental health services within broader health and social systems, the reforms aim to create holistic, seamless care pathways. The focus on sustainability and accountability underscores the importance of strong governance, financial stability and transparent monitoring.

The implementing actions in the logframes provide a glimpse into how countries planned to implement their strategies. There is a focus on capacity-building to equip health workers and leaders, expanding coverage through service integration and community-based interventions and embedding mental health support in emergency responses. Evidence-informed decision-making, resource mobilization and advocacy are key strategies to sustain service delivery, adapt policies and increase service demand. This dual focus on principles and actionable processes highlights the importance of bridging vision with implementation to achieve comprehensive, participatory reforms. The differences identified across logframes reflect country-specific needs and realities, and were expected.

### Lessons learned

Many lessons from the planning process in the nine participating countries of the WHO Special Initiative for Mental Health have emerged.

First, it was essential to create heterogeneous groups representative of the main stakeholders in the country. This supported

comprehensive and holistic planning and promoted engagement from potential future partners from the outset. Country workshops were crucial for fostering collaboration among diverse mental health stakeholders and facilitating a holistic approach to mental health system reform. Additionally, the active involvement of people with lived experiences across all countries not only enriched the logframes with valuable insights but also marked a significant step forward in recognizing and incorporating their voices in national mental health strategies.

Second, it was important to appropriately manage the breadth of priorities that mental health system reform demands. In most countries, there was a very long list of elements that needed to be worked on for mental health systems to be strengthened. Group facilitators needed to work hard to keep in mind the limits and realities of what can be achieved with the given resources and time.

Third, all plans for mental health system strengthening need to take into consideration all health system building blocks. Each health system building block is inherently tied to the others. Therefore, only focusing on the creation of services will not suffice to develop sustainable and long-lasting systems, as other related elements are inextricably linked, such as information systems, financing systems, drug management systems and human resources management systems.

Fourth, both "expert-driven" and "community-driven" approaches are important in health system strengthening. Communities, people with lived experiences, carers and families, as well as government officials, researchers, experts and international organizations all have important roles to play and can significantly contribute to mental health system transformations. This study was a starting point for the WHO Special Initiative for Mental Health to begin drawing out lessons emerging from the program. The study, therefore, has limitations. The groups responsible for planning were highly variable across country design processes, depending on whether co-creation of designs was held face-to-face or online, the number of people who could be accommodated for inputs and the capacity of country leadership (WHO and ministries of health) to facilitate discussions. Furthermore, workshop facilitators were themselves guided by health system strengthening building blocks as a framing for facilitating design workshops. Consequently, country designs and logframes tended to reflect these building blocks to some extent. Finally, the analysis of logframe data and the preparation of this manuscript were carried out by some of the workshop facilitators themselves.

## Conclusions

The WHO Special Initiative for Mental Health has demonstrated significant progress in promoting mental health system reform across nine diverse countries. This comprehensive planning effort, involving multi-stakeholder consultations and thematic analysis of country-specific logframes, revealed several common strategic priorities essential for effective mental health reform.

The study's findings can guide future mental health system reform efforts, emphasizing the need for a balanced approach that integrates expert-driven strategies with community-driven initiatives.

## Abbreviations

| | |
|---|---|
| CRPD | Convention on the rights of persons with disabilities |
| HIV/AIDS | Human immunodeficiency virus/acquired immunodeficiency syndrome |
| NGO | Non-governmental organization |
| PCMH | Philippine Council for Mental Health |
| PRIME | Program for improving mental health care |
| UHC | Universal health coverage |
| UN | United Nations |
| WHO | World Health Organization |

**Open peer review.** To view the open peer review materials for this article, please visit http://doi.org/10.1017/gmh.2025.28.

**Data availability statement.** The datasets used in this study can be accessed upon request from the corresponding author.

**Acknowledgements.** We would like to thank our colleagues Dan Chisholm and Sudipto Chatterjee for their advice and support in carrying out this research and writing this manuscript.

**Author contribution.** G.S. carried out the data analysis and was the main writer of this article. A.S. supported and supervised the data analysis process and strongly contributed to the writing of this article.

**Financial support.** No specific funding was received to carry out this research. At the global level, the WHO Special Initiative for Mental Health receives funding from the Swiss Development Cooperation agency, the Norwegian Agency for Development Cooperation and the U.S. Agency for International Development.

**Competing interests.** The authors declare none.

**Ethics statement.** This study did not require ethics approval as it did not involve human participants or human data. All research activities were conducted in accordance with the applicable guidelines and regulations of the WHO.

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
