## [Reviewer Report]

This is an important study which reports on the findings of the ambitious WHO special initiative for mental health in nine countries. The study focuses on thematic analysis of the main commonalities across the countries’ plans, following multi stakeholder workshops and the development of national level logical frameworks. The novelty of the study is its emphasis on mental health system reform through focusing on health system building blocks such as human resources, financing, governance, information management and access to medicines. The study will be of interest to other countries interested in scaling up community-based mental health services and protection of human rights. The thematic analysis and coding were conducted appropriately, and the methods of the study are well aligned with its aims.

I have a few questions/concerns:

1. A key question is whether the country implementation plans were able to attract the necessary financial support, for meaningful implementation at scale. This is not addressed in this study. While I accept that the focus of this study was on the common themes across the nine countries’ implementation plans and log frames, this would be greatly strengthened by some indication of the investment by countries (and/or other donors). This is also important for other countries to gauge what investment is required and what is possible with that investment. Financing is also a key WHO health system building block, so it doesn’t make sense to leave this out of an initiative that focuses on health systems.

2. The analysis focused on the outcome and output statements within the log frames for each of the countries but not on the activities or the indicators. This makes it difficult to understand the specific activities and indicators which were envisaged in these log frames and indeed what implementation was intended on the ground. The paper would be strengthened by including some information on these activities and indicators.

3. Similarly, although the analysis identified 5 high level themes, the authors choose to focus on only three and leave out two key themes, related to approaches and principles, and implementing actions. This leaves the reader with a very partial view of the program and what was implementation activities were envisaged. The authors do mention that this material is covered in other planned publications, but the result for the reader is a very partial perspective on this important and novel initiative. This is particularly problematic because these data are clearly available from the log frames, and could be reported in this manuscript.

4. This study identifies important commonalities across these diverse countries, but it would be useful to expand on the country specific differences in the log frames. This is briefly mentioned, but could be expanded on, to provide lessons for countries that face specific challenges. For example, were there specific challenges related to conflict affected settings or countries vulnerable to climate change, or countries with high levels of income inequality, and how were these addressed? This might provide valuable lessons for countries that may need to respond to idiosyncratic challenges – specifically on how they might respond.

---

## [Reviewer Report]

This is a paper that presents a study aiming to identify similarities and differences across Special Initiative country logframes, and to obtain results that “may assist other countries to undertake similar prioritization and planning”.

The paper is well structured and well-written. The justification and objectives of the study are well described, the methods are appropriate and correctly described, and the results are well presented.

However, with regards to the discussion and the lessons learned, I am afraid that the fact that the reform of mental health services and deinstitutionalisation are not included among the priorities that are defined by the authors is in contradiction with something that was emphasised by all countries (see lines 189-195), and is also in contradiction with the priorities repeatedly defined by WHO. Moreover, this omission could represent a dangerous signal for policy-makers looking at this paper for guidance.

As mentioned in the 2022 WHO Mental Health Report: Transforming mental health for all, "at the heart of mental health reform for most countries, lies a major reorganization of mental health services. The task is to simultaneously shift the locus of care for severe mental health conditions away from any institutions and towards communities, while scaling up the availability of care for common conditions such as depression and anxiety. Both strategies are critical to advance human rights and improve the coverage and quality of mental health care. Every district, province, prefecture, region, major city or other sizeable administrative division (here called “district”) should have a network of accessible community-based mental health services to provide an inter-connected platform for supporting people with a broad range of mental health conditions”.

It is true that, as the authors say, “only focusing on the creation of services will not suffice to develop sustainable and long-last systems, as other related elements are inextricably linked, such as information systems, financing systems, drug management systems and human resources management systems”. However, the development of sustainable and long-lasting systems with the capacity to respond to the mental health care needs of the population is not possible without replacing the psychiatric hospitals with community based services including mental health care in primary care, inpatient units in general hospitals, community mental health teams, psychosocial rehabilitation programmes and small-scale residential facilities, among others. All this constitute a very complex process that can only be successful if policy makers see it as one of the main priorities in mental health, especially in countries that spend the large majority of their mental health budget in psychiatric hospitals. If this reform is not a priority, it will never be possible to develop community based care and increase human rights of persons with mental disabilities. For all these reasons, I would suggest the authors to include mental health services reform and deinstitutionalisation among the priorities in mental health.

---

## [Reviewer Report]

Thank you for the careful consideration of my earlier comments. i’m happy with the changes that you have made.

---

## [Editor Report]

Thank you for amending your paper as per reviewers‘ suggestions. The paper seems okay now. Please include impact statement and make sure that the journals’ author guideliness are fully complied to.